# Paleocene/Eocene carbon feedbacks triggered by volcanic activity

Sev Kender [1,2✉], Kara Bogus[1], Gunver K. Pedersen [3], Karen Dybkjær[3], Tamsin A. Mather [4], Erica Mariani [1], Andy Ridgwell [5], James B. Riding[2], Thomas Wagner [6], Stephen P. Hesselbo [1] & Melanie J. Leng[7]

The Paleocene–Eocene Thermal Maximum (PETM) was a period of geologically-rapid carbon release and global warming ~56 million years ago. Although modelling, outcrop and proxy records suggest volcanic carbon release occurred, it has not yet been possible to identify the PETM trigger, or if multiple reservoirs of carbon were involved. Here we report elevated levels of mercury relative to organic carbon—a proxy for volcanism—directly preceding and within the early PETM from two North Sea sedimentary cores, signifying pulsed volcanism from the North Atlantic Igneous Province likely provided the trigger and subsequently sustained elevated $CO_2$. However, the PETM onset coincides with a mercury low, suggesting at least one other carbon reservoir released significant greenhouse gases in response to initial warming. Our results support the existence of 'tipping points' in the Earth system, which can trigger release of additional carbon reservoirs and drive Earth's climate into a hotter state.

[1] Camborne School of Mines, University of Exeter, Cornwall, UK. [2] British Geological Survey, Nottingham, UK. [3] Geological Survey of Denmark and Greenland (GEUS), Copenhagen, Denmark. [4] Department of Earth Sciences, University of Oxford, Oxford, UK. [5] Department of Earth and Planetary Sciences, University of California at Riverside, Riverside, CA, USA. [6] Lyell Centre, Heriot-Watt University, Edinburgh, UK. [7] National Environmental Isotope Facility, British Geological Survey, Nottingham, UK. ✉email: s.kender@exeter.ac.uk

The relative geological rapidity of warming and $CO_2$ release at the Paleocene–Eocene Thermal Maximum (PETM), and the potential activation of feedbacks between warming and organic carbon reservoirs ~56 million years ago[1], have relevance to understanding future Earth system responses to ongoing anthropogenic perturbation[2]. However, the major sources of carbon and the causal mechanisms triggering its release have remained under debate[3–7], stymying our ability to draw firm inferences relevant to the future. Plausible carbon sources include peatlands and permafrost[3], methane hydrates[8], sedimentary marine organic matter[9,10], and the mantle[5], while proposed hypotheses for the PETM trigger include changes in orbital insolation[1,3,11], volcanic activity of the North Atlantic Igneous Province (NAIP, Fig. 1)[12,13], and an extra-terrestrial impact[14]. The problem has been in de-convolving the possible multiple different sources of carbon that contributed to the sharp decline in carbon isotope ($\delta^{13}$C) values observed in the sedimentary record marking the PETM onset and importantly, separating triggers from carbon-climate feedbacks. For example, a recent study[5] concluded that volcanically produced carbon was the main source to explain proxy records of ocean pH across the PETM, but was unable to resolve the relative timing and contribution of different kinds of volcanic and non-volcanic carbon. Other studies have shown that NAIP volcanic rocks (e.g., lava flows, ash beds and sills) were formed approximately around the time of the PETM[4,10,12], but limited dating of individual local geological features makes it difficult to conclude confidently whether volcanism provided the trigger. Here, we investigate new sediment cores from the North Sea for both high resolution $\delta^{13}$C stratigraphy and sedimentary mercury (Hg) as a proxy for volcanic emissions.

Variation in the concentration of Hg in organic carbon-rich marine sediments has shown promise as a direct means of elucidating regional-scale volcanic activity such as associated with the PETM[7,15,16] as well as earlier large igneous provinces[17]. Today, approximately 20–40% of natural global Hg emissions come from the Earth's crust via volcanism[18]. Released to the atmosphere, the ~0.5–1 year residence time of Hg allows it to be mixed globally, and when subsequently deposited to the ocean or as peatlands its preservation is predominantly via association with organic matter[19–23]. As such, normalised to total organic carbon (TOC), Hg has previously been used as a proxy for global volcanism[17]. Mercury can also be released hydrothermally, with modern hydrothermal vents being associated with locally-elevated Hg concentrations in water, sediments, and biota[24]. In the case of the PETM, Hg released via hydrothermal vents[4,12] associated with the NAIP may not necessarily have entered the atmosphere but instead be detectable in NAIP-proximal ocean sediments[7], where the modern residence time of Hg in the open ocean is decades to centuries, and on shelves is ~4 months which is considerably less than the residence time of shelf water[25].

To constrain the timing of pulsed volcanism across the PETM and hence help elucidate its role as a trigger, we present the first high-resolution records of Hg from sediment cores 22/10a−4 (ref. [26]) and E−8X (~100–500 years per sample; Figs. 1 and 2). Although modern dissolved Hg input from freshwater runoff to oceans is substantial ($27 \pm 13$ Mmol year$^{-1}$)[27] compared to atmospheric deposition ($19.5 \pm 9.5$ Mmol year$^{-1}$)[28], isotopic studies have shown terrestrial Hg to be constrained to coastal and nearshore shelf environments[29] whilst our core sites are >200 km distal in the central North Sea. Well sites 22/10a−4 and E−8X are

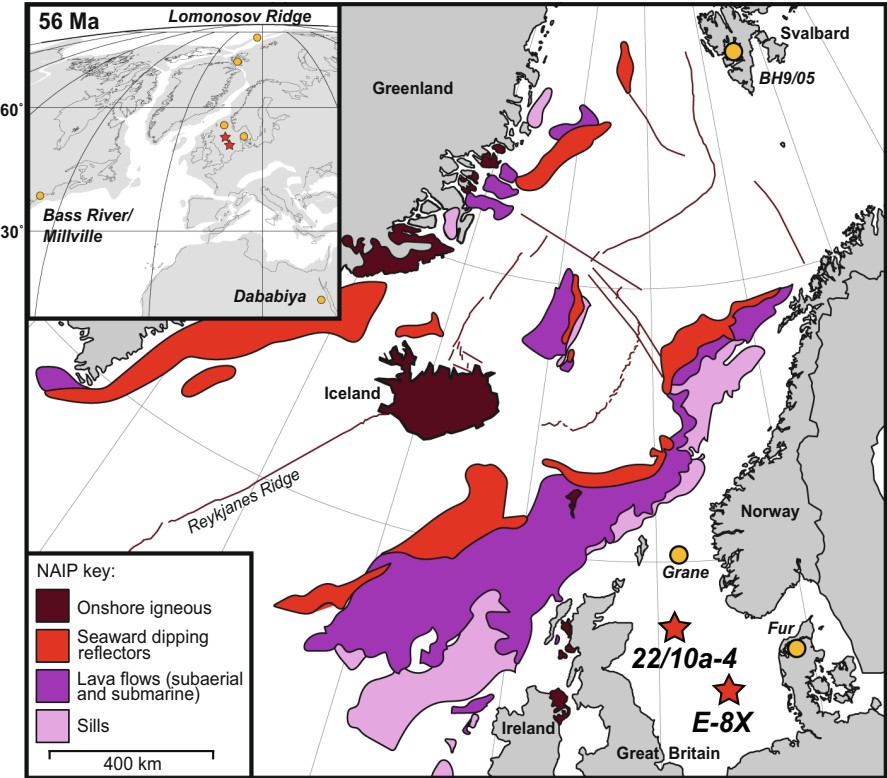

**Fig. 1 Location maps of the North Atlantic Igneous Province (NAIP) and sediment cores sites analysed in this study.** The simplified NAIP main map shows the estimated ranges of its various components[61]. 'Seaward dipping reflectors' are well-defined seismic reflectors beneath the uppermost basalt, interpreted as large subaerial sheet lava flows associated with rifting[61]. Other lava flows are thought to be a combination of subaerial and submarine, and sills were considered as intruded into the upper crust[4,12,61]. The insert map is a Mollweid projection of modern continents (lines) on a palaeogeographic reconstruction, generated from (ref. [62]), of continental plates (grey) centred at 56 Ma.

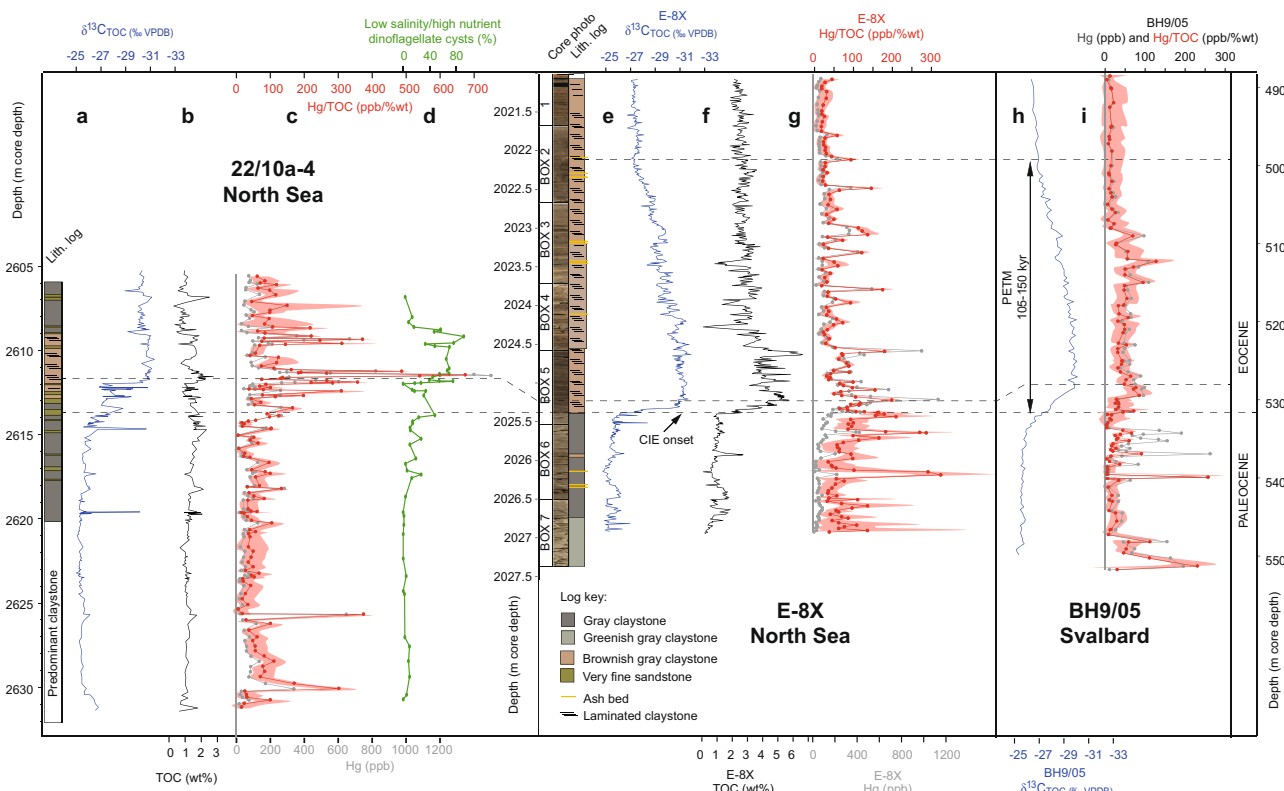

**Fig. 2 Geochemical proxy records from the North Sea and Svalbard cores, showing volcanic and sedimentological changes associated with the Paleocene–Eocene Thermal Maximum (PETM).** Records are shown against depth in m core depth (below oil rig floor 'Kelly bushing' for 22/10a-4 and E−8X). **a–d** Well site 22/10a-4 (North Sea). **e–g** Well site E−8X (North Sea). **h–i** Core BH9/05 (Svalbard). Bulk sediment total organic carbon δ13C$_{TOC}$ is reported as ‰ VPDB, Vienna PeeDee Belemnite. Total organic carbon (TOC) is reported as % of the bulk weight. Hg is reported as parts per billion (ppb). Hg/TOC envelope reflects an analytical error, illustrating higher uncertainty in samples with lower TOC. The 22/10a-4 lithological (lith.) log, δ13C$_{TOC}$, and TOC from are from (ref. [26]). The BH9/05 δ13C$_{TOC}$ and age model are from (ref. [40]), and Hg data are from (ref. [7]). The position of the Paleocene/Eocene boundary, defined as the onset of the PETM, is shown as a horizontal dashed line.

located ~400 km and ~600 km proximal to NAIP volcanic centres, respectively (Fig. 1). These sediment cores contain elevated levels of organic carbon suitable for this analysis[7,30] and are from deep cored material that has not undergone weathering, a process that has been shown to change the Hg signal in some outcrop samples[30]. To constrain the PETM onset in detail, we also present a new and exceptionally well-defined δ13C and TOC record for core E−8X and use this to correlate between well sites and to create a consistent age model with orbitally-tuned Svalbard core BH9/05 (Fig. 3 and "Methods").

Here, we document numerous peaks in Hg and Hg/TOC at both sites E−8X and 22/10a−4 above background, which occur both immediately before and within the PETM. By comparing with records elsewhere, we interpret these as evidence of episodic NAIP sill emplacement releasing thermogenic methane and Hg into ocean water via hydrothermal venting. Our Hg records provide evidence that the onset of the PETM was triggered by volcanic activity, but we find at least one other carbon reservoir must have subsequently been released as Hg and Hg/TOC decline in the second part of the PETM onset.

## Results

**Mercury pulses over the PETM.** Both cores E−8X and 22/10a−4 show numerous high frequency and high amplitude fluctuations in Hg and Hg/TOC above background levels of ~50 ppb and ~40 ppb/wt%, respectively (Figs. 2 and 4). Modern oceanic sedimentary Hg has not been comprehensively constrained, and in many settings is thought to be contaminated by anthropogenic

input[22,31–33]. Recent work has shown the Mediterranean Sea seafloor sediments from ~0.2 to 4 km water depth contain average Hg concentrations of 66 ppb and Hg/TOC values of 133 ppb/wt%, with isotopic modelling suggesting ~75% of the Hg may be from urban or industrial pollution[33]. Perhaps more comparable to the Paleogene North Sea, Baltic Sea sediments show average Hg concentrations of 20–40 ppb, and Hg/TOC of ~15 ppb/wt%, for preindustrial sediments[32]. Average shale Hg over the Phanerozoic has been estimated as 62 ppb, and average Hg/TOC as 71.9 ppb/wt% (ref. [34]), although these datasets are skewed towards studies of Large Igneous Province volcanism and may therefore be above overall average shale background. In contrast, our records show Hg spikes up to values comparable to modern contaminated sediments of >1000 ppb (ref. [35]), and Hg/TOC spikes of 200–500 ppb/wt%. Looking further afield to Svalbard (Fig. 1), marginal marine core BH9/05 has lower concentrations of Hg (ref. [7]), but also shows numerous Hg/TOC pulses (Fig. 2). Although the Hg pulses within these cores may not quantitatively constrain the extent of volcanism, as eruptions may produce varying quantities of Hg (ref. [18]) and sedimentary transport/preservation effects might also come into play, submarine eruptions would likely produce predominantly local Hg enrichments[36]. Any occurrence of marine anoxia cannot account for the Hg, as increased sedimentary drawdown would have rapidly depleted the water column if dissolved Hg were not continuously replenished[17]. Neither can Hg released from permafrost melting account for the observed (up to a factor of 4) Hg increases, as releasing even a modern permafrost inventory of

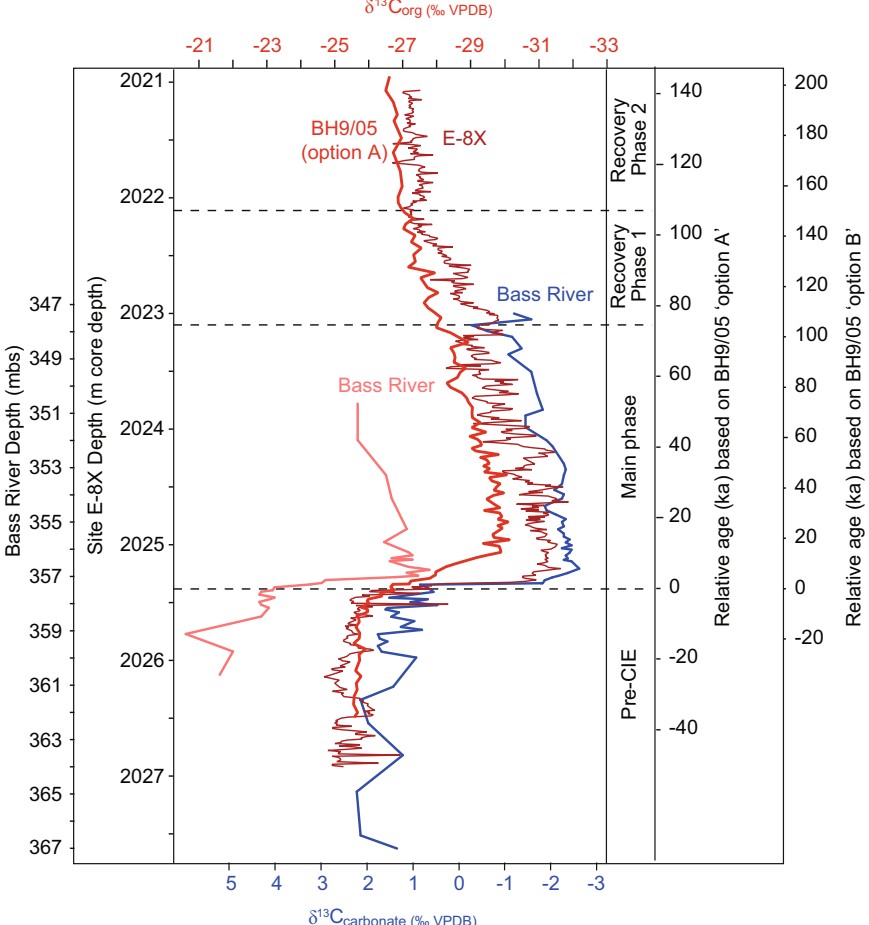

**Fig. 3 Carbon isotope correlation of two sites with Svalbard core BH9/05.** The $\delta^{13}C$ of both organic carbon ($\delta^{13}C_{org}$) and inorganic carbonate ($\delta^{13}C_{carbonate}$) from North Sea well site E−8X (this study) and Bass River[38], are correlated to Svalbard core BH9/05 (ref. [40]) based on the overall shape of the records, with particular emphasis on the carbon isotope excursion (CIE) inflection points during the rapid onset and gradual recovery phases. The relative age model is based on two proposed solutions for cyclostratigraphy of core BH9/05 (ref. [40]). Bass River core depth in metres below the surface (mbs).

330–1300 Gg of Hg (ref. [23]) over 2 kyear (our estimated approximate Hg pulse duration) equates to a release rate no more than the modern volcanic background rate of 700 Mg Hg/year (ref. [18]), with this doubling unlikely to register clearly in the composition of sediments given other causes of background fluctuations. Finally, our records are characterised by multiple peaks of Hg/TOC, inconsistent with a single bolide impact or permafrost melting event[16]. Therefore, our results do suggest pulsed volcanic Hg releases occurred sporadically throughout the study interval[7,37], and in particular around the PETM onset.

Core 22/10a−4 has the longer pre-PETM section (Fig. 2c) and records a probable early phase of volcanism with two elevated Hg and Hg/TOC pulses between 2630 and 2625 m. Although we cannot with any certainty correlate these with other sites to assess if the feature was of local or regional scale, there are early pulses at cores E−8X and BH9/05 (Fig. 2g and i). Pulses in sedimentary Hg and Hg/TOC increase in the lead up to the PETM onset, in particular in E−8X (~2025.6 m) and BH9/05 (~535 m), but also at 22/10a−4 (~2617 m). At the onset of the carbon isotope excursion (CIE; defining the onset of the PETM), all three sites show a pulse in Hg and Hg/TOC (dashed line marked 'CIE onset' in Fig. 2), and then further pulses encompassing the CIE onset and main phase. Hg/TOC pulses peak in 22/10a−4 during the earliest CIE phase. A broader regional—rather than local—

volcanic source for these pulses is indicated by their presence in E−8X and 22/10a−4 ~350 km apart, and in BH9/05 at Svalbard (although the spikes are less pronounced). Hg/TOC pulses encompass the onset of laminated sediment (likely from the removal of bioturbating organisms due to inhospitable bottom water conditions), and surface ocean eutrophication/lower salinity of the North Sea inferred from dinoflagellate cysts[26] (Fig. 2d), although they do not correlate with these proxies. Hg/TOC pulses decline in frequency and amplitude at both North Sea sites as $\delta^{13}C$ begins to recover, approximately coinciding with the end of deposition of laminated sediments and shift away from eutrophic/low salinity surface water conditions at 22/10a−4 (~2608 m), and a shift towards lower TOC values at E−8X (~2024.5 m). Further Hg pulses occur at E−8X and BH9/05 and then decline further as $\delta^{13}C$ continues to fully recover.

**Carbon cycle and mercury changes.** The timing of Hg/TOC pulses relative to the evolution of $\delta^{13}C$ is consistent with volcanism triggering PETM carbon release and also contributing to sustaining the CIE during its main phase[4,12]. This is evidenced by the Hg and Hg/TOC peaks immediately before and during the initial negative CIE onset, and later within the main phase of the PETM before $\delta^{13}C$ begins to increase again (Fig. 2). We assume that our proxies for surface ocean $CO_2$ and volcanism (paired $\delta^{13}C$ and Hg/TOC data, respectively) are essentially synchronous

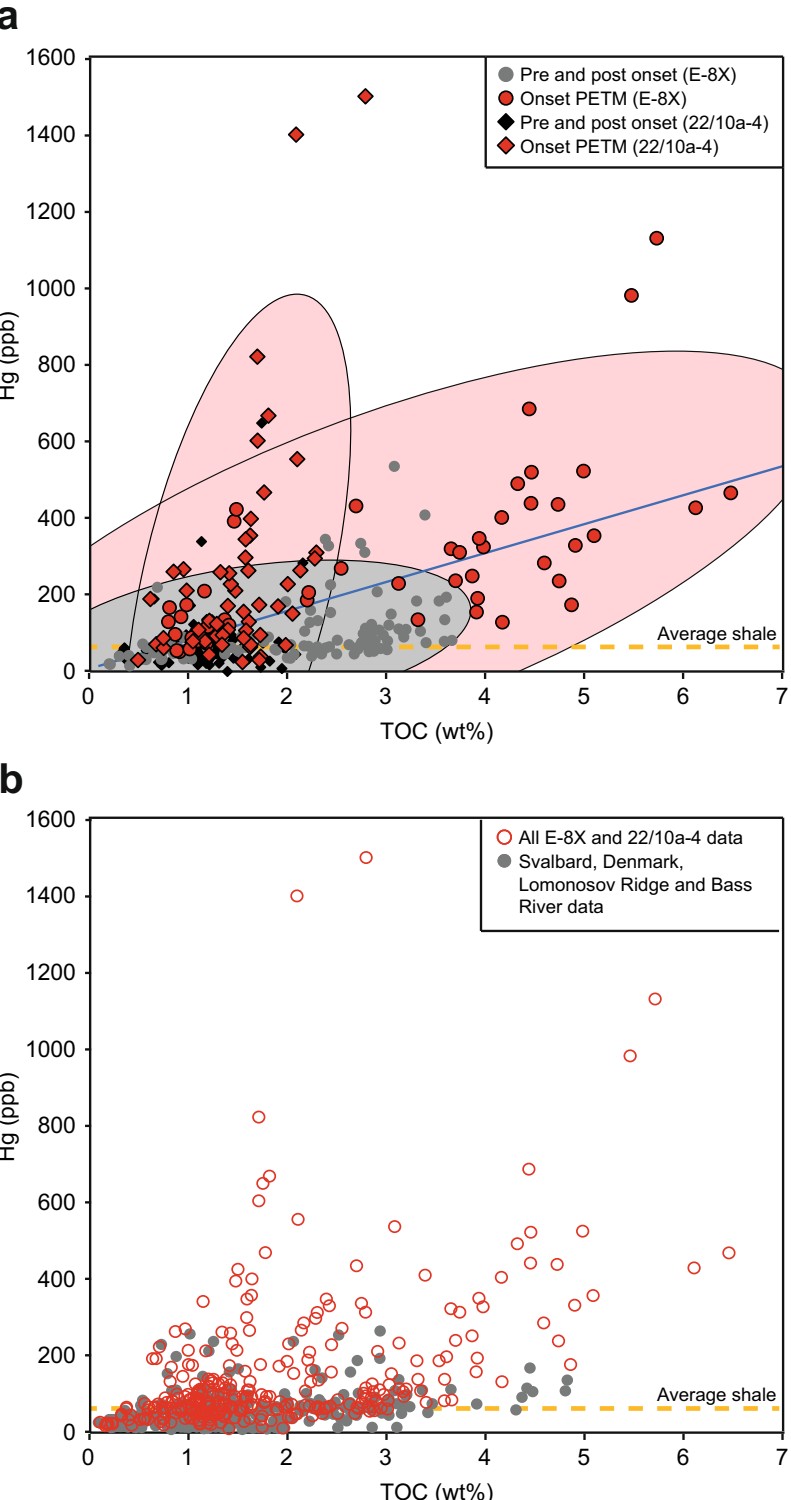

**Fig. 4 Sedimentary Hg (ppb) against TOC (wt%) for well sites E−8X and 22/10a−4 and other published records. a** Line is the linear regression of E −8X and 22/10a−4 datasets combined ($R^2 = 0.22$). Samples immediately before, during and after the Paleocene–Eocene Thermal Maximum (PETM) onset are indicated as red symbols (~2024.6–2026 m for E−8X; ~2608–2615 m for 22/10a−4), and outside of that area with grey symbols. Shaded 95% ellipses show the changing relationship between Hg and TOC over the studied interval, with many samples within the PETM onset (red symbols) exhibiting excess Hg with a steeper gradient to TOC than samples outside of this interval (grey symbols). Well site 22/10a−4 appears to have experienced greater excess Hg than E−8X, possibly as it was closer to the North Atlantic Igneous Province source. The dashed line shows the value of the average Phanerozoic bulk shale[34]. **b** All data from E−8X and 22/10a−4 (red symbols) plotted with data from Svalbard BH9/05, Denmark Fur formation, Lomonosov Ridge and Bass River (grey symbols)[7].

at our (continuous at E−8X) sampling resolution (>100 years per sample). This is because the oceanic residence time of shelf Hg, and the time it would take volcanic $CO_2$ to reach the atmosphere, mix, and be sequestered into ocean sediment via phytoplankton, is months to years. These Hg/TOC pulses are not likely to have been caused by changing sedimentation rates, as reporting Hg as a ratio to TOC aims to remove the influence of changing back-ground sedimentation. However, changing delivery and source of sedimentary TOC (transportation and reworking) can modify Hg/TOC trends. Well sites E−8X and 22/10a−4 were within the deepest part of the North Sea Basin ~200 km from land, at palaeo-water depths of up to ~500 m, well below storm wave base[26]. Well site E−8X is exceptionally useful for Hg analysis as we see no evidence for highly variable sedimentation rates in the high TOC fine-grained claystone. The location and sedimentation suggest the carbon was at least partly of marine origin, as the $\delta^{13}C$ record shows a similar CIE shift to that measured on dino-flagellate cysts at New Jersey[38], and shows no smoothing that might typify a transported terrestrial and/or reworked carbon setting[39].

## Discussion

Well sites E−8X and 22/10a−4 records show broad similarities in Hg/TOC, but transported carbon could partly explain why the Hg/TOC signals from Svalbard[7] are somewhat different (Fig. 2). Our $\delta^{13}C$ correlation illustrates that core BH9/05 appears to have a more extended CIE onset[40] than at E−8X, Fur Island (Denmark), or Bass River (New Jersey) (Figs. 3 and 5). This is most likely due to its marginal marine nature and proximity to land (evidenced by palynology changes[40]) which is argued to have exposed that site to significant reworked terrestrial carbon that could have residence times of thousands of years[39], possibly muting some Hg signals. However, records from Svalbard and Denmark are useful for constraining the extent of Hg signals, and broadly show a baseline decrease in Hg/TOC and Hg with distance from the NAIP (Figs. 4 and 5). These signals together imply that volcanically-sourced Hg during the PETM may have been largely released into ocean water

proximal to the NAIP, supporting the hypothesis of hydrothermal venting of thermogenic carbon from volcanically-emplaced sills[4,12]. Although the most NAIP proximal site Grane (Fig. 1) has by far the highest reported Hg values, it has a poorly defined carbon isotope stratigraphy, and exceptionally high Hg values may be related to local hydrocarbon migration and overprinting[7]. Similarly, although the most distal PETM Hg records exist from outcrops in Egypt[15], interpretation is hampered by low TOC and severe weathering and dissolution that has been shown to alter primary Hg signals[30]. The lowest observed Hg comes from distal PETM sites at New Jersey[7,16] and Blake Nose[16] consistent with NAIP activity releasing Hg largely into proximal sediments, although TOC values are often below the analytical precision required for robust Hg/TOC assessment[7].

The high resolution and clear carbon isotope signals from core E-8X allow examination of the structure of the CIE onset, where the negative $\delta^{13}C$ shift takes place over two steps of persistent 1.5–2‰ decrease; CIE step 1 and CIE step 2, each lasting in the region of 1.5–2 kyear (Fig. 6c). Although CIE step 1 is associated with a Hg/TOC pulse above the background (Fig. 6a), CIE step 2 is not, even though step 2 represents the largest change in $\delta^{13}C$ from presumed atmosphere-ocean carbon release. The onset of the Hg/TOC pulse immediately before CIE step 1 (~2025.42 m; just before time 0, Fig. 6a) coincides with a slight increase in Hg (from 99 to 167 ppb) and a slight reduction in TOC (from 1.1 to 0.8%). Although the elevated Hg/TOC values are therefore being partly driven by reducing TOC as well as increasing Hg, values are still well above analytical uncertainty (red shading in Fig. 6a). It is unlikely that poor preservation of TOC at this horizon caused the initial pulse, and the Hg/TOC remains high into the start of step 1 when TOC increases to ~2%, supporting that this pre-CIE step 1 Hg/TOC elevation is at least partly indicating elevated volcanic activity rather than entirely explainable as due to changes in TOC preservation, delivery, or development of anoxia. Indeed, Hg/TOC pulses continue to interrupt the record and are not systematically coupled with TOC. We, therefore, suggest $CO_2$ emissions driving CIE step 1 were at least partly from hydro-thermal venting and volcanic sill emplacement (due to elevated

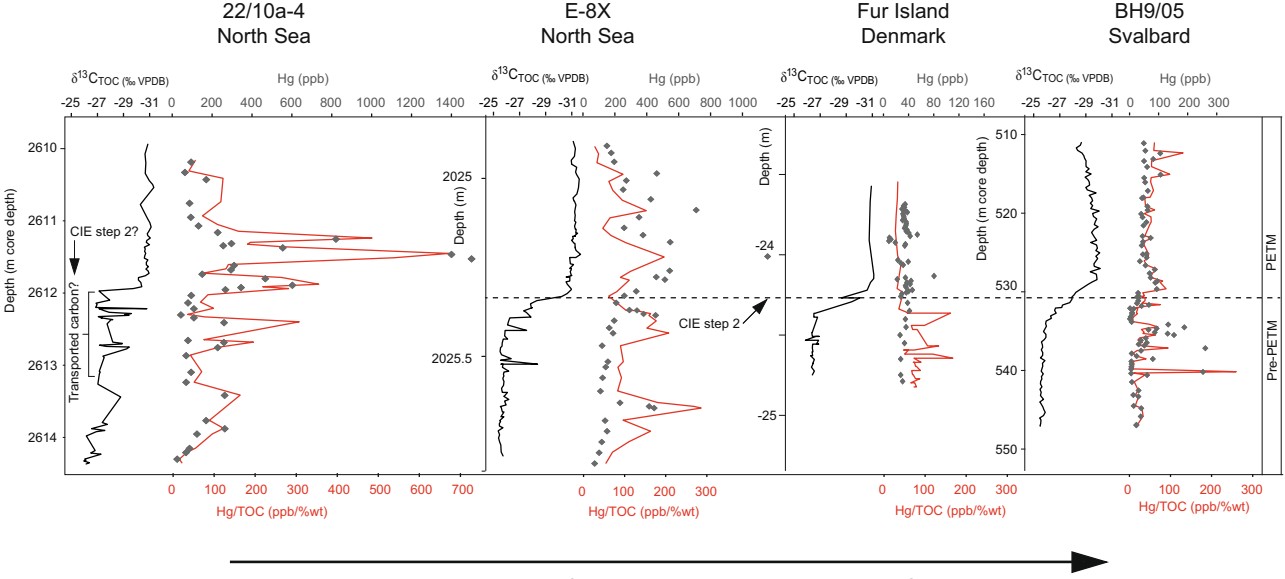

**Fig. 5 Summary of Hg and Hg/total organic carbon (TOC) data from various sites at the onset of the PETM.** North Sea well sites 22/10a−4 and E−8X (this study) generally display higher values than Fur and Svalbard[7]. Carbon isotope excursion (CIE) step 2 is shown as a dashed line and does not co-occur with a Hg or Hg/TOC spike in the sections. Core 22/10a−4 has previously been interpreted to have been partially impacted by transported carbon[26]. Bulk sediment $\delta^{13}C_{TOC}$ is reported as ‰ VPDB, Vienna PeeDee Belemnite.

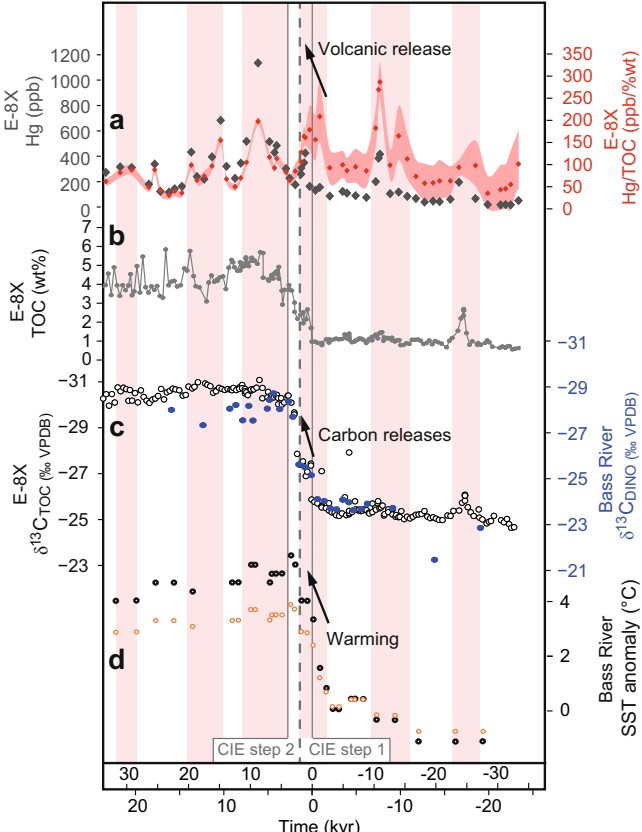

**Fig. 6 Proxies for volcanism, carbon release and temperature in the time domain; thousands of years from the start of the PETM carbon isotope excursion (CIE). a** Sedimentary mercury (Hg) and Hg/total organic carbon (Hg/TOC) for core E−8X (this study). Hg/TOC envelope reflects an analytical error, illustrating higher uncertainty in samples with lower TOC. **b** Sedimentary TOC weight (wt) % for core E−8X (this study). **c** Sediment proxy records for TOC isotopes ($\delta^{13}C_{TOC}$) at E−8X (this study), and carbon isotopes of sedimentary organic-walled dinoflagellate cysts ($\delta^{13}C_{DINO}$) from Bass River, New Jersey[38], used to correlate the two sites (Fig. 3). **d** Sea surface temperature (SST) anomaly proxy data from Bass River, New Jersey[38]. The same data are presented with two different calibrations for temperature in orange[63] and black[64].

Hg/TOC in the North Sea), although we acknowledge our records cannot discount potential additional sources of $CO_2$. The latter part of CIE step 1 coincides with increased Hg (Fig. 6a), before both Hg and Hg/TOC reduce into CIE step 2, potentially evidencing a lull in volcanic activity. This drop-in Hg/TOC during CIE step 2 can also be seen in BH9/05 (Svalbard), Fur Island (Denmark), and possibly 22/10a−4 (Fig. 5), although the latter site contains some noise in the $\delta^{13}C$ record-making relationships less clear. Some records are thought to be influenced by sediment reworking (e.g., Svalbard and 22/10a−4) that could reduce Hg/TOC signals, but E−8X shows no sign of reworking with a clear and rapid CIE onset and central deep North Sea Basin location. In Svalbard, there is a general increase in the Hg/TOC baseline over the CIE onset, and at Fur sedimentation rates significantly increase during the PETM[41] such that Hg deposition rates likely increase even though concentrations do not. However, there is no Hg/TOC evidence for any substantial increase in volcanism during CIE step 2 (unlike step 1). We note that all other E-8X instances of reduced Hg/TOC in this interval (vertical white bars in Fig. 6) coincide with increasing $\delta^{13}C$ (and presumed $CO_2$ drawdown), and increases (pink bars in Fig. 6) coincide with

decreasing $\delta^{13}C$ (and presumed $CO_2$ releases), although the changes outside of the CIE steps are subtle.

Although we do not rule out background volcanic activity occurring during CIE step 2, if volcanism was voluminous enough to produce the required $CO_2$, annual release rates would have been an order of magnitude above modern volcanic $CO_2$ emissions, and therefore possibly raised annual Hg deposition to at least an order of magnitude above modern[18] and be detected in our samples. Alternatively, if intruded into organic-rich mudrocks such as those that underlay much of the NAIP[12] we might expect even higher Hg deposition in the North Sea sediments. Therefore, this short-lived reduction in Hg/TOC during CIE step 2 is important as it points to a secondary phase of carbon release from a reservoir not directly linked to Hg emissions and hence likely not to volcanism. The main possibilities for such a feedback reservoir are suggested to include methane hydrates[8] and permafrost carbon[3].

While other pulses of Hg/TOC at E−8X do not correlate with such a large decrease in $\delta^{13}C$ as CIE step 1, they do consistently co-occur with modest decreases in $\delta^{13}C$ both before and after the CIE onset (pink bars in Fig. 6) which may signify thermogenic $CO_2$ releases. Interestingly, previous studies including at Bass River have documented precursor environmental changes to the PETM which include sea surface temperature (SST) increase[38,42]. We correlate E−8X $\delta^{13}C$ with Bass River dinoflagellate cyst $\delta^{13}C_{DINO}$, which also shows the two-step CIE onset (Fig. 6c), and find that this early warming (Fig. 6d) corresponds with Hg/TOC evidence for volcanism which we speculate may have been its cause. We note that SST warming began even earlier, at about 10 kyear before the CIE onset (Fig. 6d), which also correlates with a Hg and Hg/TOC spike in E−8X although we recognize that correlation between Bass River and the North Sea outside of the CIE onset interval is less certain. SST records from nearby Fur[43,44] are slightly more complicated to interpret due to occasionally high branched and isoprenoid tetraether (BIT) index values and changing sedimentation rates making it harder to correlate, but do show a possible fall stratigraphically below the CIE that has been suggested to reflect local cooling from volcanism[44].

The coincidence of the largest global shift in $\delta^{13}C$ (CIE step 2; Fig. 6) with reduced volcanic activity (suggested by reduced Hg/TOC; Figs. 5 and 6), points to the activation of a secondary, unstable/labile carbon reservoir that was depleted in response to initial warming possibly from NAIP volcanism. Although the Eocene is not directly analogous to Earth's current markedly cooler climate state, our records are consistent with a tipping point whereby an additional warming-driven carbon release pushed the world into the PETM 'Hothouse Earth'. Comparable processes have been predicted for the future if significant mitigations are not carried out[45]. We acknowledge that numerous uncertainties remain in the construction of our age model, and modelling is now needed to assess the likely source of secondary carbon release and estimate the amount of warming and volume of greenhouse gasses emitted. Additional records and proxy constraints are needed to confirm our diagnosed transition from volcanism-dominated to climate feedback-dominated carbon release at the onset of PETM, and the global warming threshold at which it occurred. However, our work highlights the utility of the palaeo record in better understanding the existence and sensitivity of carbon-climate feedbacks and potential tipping points.

## Methods

**Core handling and sampling.** Core from well site E-8X (55°38′13.42″N; 04°59′11.96″E; Supplementary Fig. S1) was drilled in 1994 for hydrocarbon exploration. Cores 3 and 4 represent most of the Paleocene succession (the Våle Formation,

Lista Formation and the lower Sele Formation; Supplementary Fig. S2). Core 3 was taken from 2021.065 m (below Kelly bushing) downwards (Supplementary Fig. S2) and cut into ~1 m sections (here termed Boxes). The upper part (Boxes 1–7) were split for the first time in 2013, after consolidation in foam and plastic tubing, with a split offset from the maximum diameter of the core. The larger part of the core (about two-thirds) was assigned as the archive, photographed (Fig. 2), and logged (Supplementary Fig. S2). The smaller part of the core was cut into two. One of these slices was continuously sampled at 1–2 cm intervals (dependent on the consolidation of the rock) and completely depleted. Sampling was carried out with a clean metal spatula, and the material was placed in labelled plastic sample bags. All analyses were performed on subsamples. Samples were labelled based on the depth from the top of each box (0 cm = top of a box). All samples were collected in a responsible manner in accordance with local laws.

**Sedimentology**. Sedimentological logs for well site E−8X (water depth 44 m) are presented in Supplementary Fig. S2 for Core 3, Boxes 1–11, 2030–2021 m, which represents the upper part of the Lista Formation (i.e., the top of the Ve Member and the Bue Member), and the lower part of the Sele Formation. The following descriptions add details to the more general lithological descriptions already described[46].

For the Ve Member of the Lista Formation, only the uppermost 1.5 m of the Ve Member (2030.03–2029.55 m) is shown on the sedimentological log and is dominated by red, homogeneous mudstone. The lithology is consolidated mud, compacted by the overlying 2 km of sediments. The Ve Member is rich in smectite[46], the preservation of which also indicates the mud was never buried to temperatures where smectite recrystallizes (>90 °C). The dominant facies of the Ve Member is homogeneous red claystone with small (mm-scale) white concretions of unknown mineralogy. Small green patches are locally present in the dark red clay. Two 15–20 cm thick beds with graded bedding, and a small slump fold near the base, occur below 2030 m. The muddy matrix contains quartz sand and white spheres (<1 mm in diameter), tentatively interpreted as redeposited concretions. The graded beds are tentatively interpreted as gravity flows. The boundary towards the Bue Member is transitional, and the uppermost 20 cm of the Ve Member are green or variegated, and weakly laminated. The Ve Member is correlated with the Holmehus Formation, onshore Denmark, and the latter has been interpreted as fully marine clay, deposited very slowly at water depths of >500 m (ref. [47]). The slow sedimentation rates may partly explain the oxidation of the marine clay.

The Bue Member of the Lista Formation is represented by the interval 2029.55–2025.48 m (Supplementary Fig. S2). The basal part of the Bue Member is characterised by variegated clay, with an irregular transition from the dark red clay of the underlying Ve Member, through pale red and pale greenish clay, to pale greenish clay at the top. The colour changes follow neither weak lamination nor distinct burrows. The TOC shows a distinct increase from 1% (2025.41 m) to 4% (2025.31 m), which marks the boundary between the Lista and Sele Formations. Two sedimentary facies are characteristic of the Bue Member in E−8X. The lower part is dominated by pale green to greyish green clayey mudstone without laminations but occasionally with trace fossils (*Zoophycos* sp. and *Chondrites* sp.). The *Zoophycos* burrows are ~2 mm in diameter, and the *Chondrites* burrows are ~1 mm. The ichnogenera *Chondrites* and *Zoophycos* are common in depositional environments with dysoxic conditions[48]. These trace fossils are known to form a deep tier below the sea floor[48–51], and it is possible that other, shallow-burrowing organisms produced the nearly homogeneous clayey mudstone. Pyrite is observed in the bioturbated sediment, either as small concretions or as pyritic laminae. In the upper part of the Bue Member (>2026.85 m), the lithology changes gradually to a greenish-black mudstone where neither laminations nor burrows are observed. Upwards, the mudstone becomes more greyish and locally fissile. The mudstone has no visible pyrite and no visible trace fossils. Shallow burrows in a 'soup ground' sediment may not be preserved as distinct trace fossils[51]. The mudstone contains few layers of volcanic ash. Towards the Sele Formation, the Bue Member is a weakly laminated, greyish black claystone or clayey mudstone, locally slightly greenish with jarosite. The TOC content increases in these upper facies (Fig. 2, Box 6), supporting the interpretation of a change from dysoxic to nearly anoxic conditions. The weak lamination suggests that only a sparse benthic fauna may have existed during the deposition of the upper part of the Bue Member. The presence of the ichnogenera *Chondrites* and *Zoophycos* suggests it may have been deposited in a dysoxic environment and that only a relatively small drop in dissolved oxygen was required to obliterate most of the benthos and thus preserve the lamination in the sediment.

The Sele Formation is ~11 m thick in E−8X, and the lower 4.3 m from the top of Core 3 (2025.3–2021 m) (Supplementary Fig. S2). The Sele Formation is dominated by dark, brownish grey to black mudstone with mm and sub-mm scale laminae interbedded with very few and thin layers of volcanic ash. The mudstone locally contains laminae enriched in pyrite. The brownish grey, laminated mudstone is interbedded with a cream coloured mudstone with 0.1 mm laminae (2023.18–2024.58 m). The paler beds contain more silt-sized particles than the background mudstone. The boundary to the underlying Lista Formation (Bue Member) is sharp. This is also supported by the TOC analyses, which show a dramatic rise in TOC across the boundary (Fig. 2). Upwards through the Sele Formation TOC decreases. The return to normal marine values is not observed in the E−8X core. The very thin, parallel and continuous laminae in the Sele Formation indicate a complete absence of benthic fauna. Anoxic conditions, with

bottom water containing lethal H$_2$S, would preclude benthic fauna, and the lack of dissolved oxygen would lower the rate of bacterial degradation of organic particles, and thus favour the development of an organic-rich mudstone. Sulphate-reducing bacteria might locally produce relatively large amounts of pyrite. Fine bodies of framboidal pyrite may be present in the mudstone. The Sele Formation is interpreted as deposited under anoxic conditions. This is also the case for the diatomaceous Fur Formation in onshore Denmark[52].

The laminated, anoxic mudstones of the Sele Formation are also known from other wells in the North Sea[46]. Onshore Denmark, dark grey, laminated clay (the Stolle Klint Clay) was deposited from the onset of PETM, which demonstrates that anoxic conditions were regional in the North Sea Basin during the PETM[41,53].

Sedimentation rates may be estimated from laminated sediments if the stacked laminae are nearly identical and deposited annually (as varves). Laminated sediment from 22/10a-4 (North Sea) and the Stolle Klint Clay (Denmark) was studied in a thin section and suggested to have been annually deposited (Kender et al. [26]; Heilmann-Clausen et al. [53]). Core 22/10a−4 was found to include average ~0.08 mm-thick couplets of pale and dark laminae, and the Stolle Klint Clay ~0.25 mm-thick couplets. If annual, the ~3.5 m-thick laminated part of 22/10a−4 would have been deposited in ~40 kyear, and the 24.4 m-thick Stolle Klint Clay[7] deposited in ~100 kyear.

**Depth and age scales**. There are three depth scales for E−8X (all m below Kelly bushing). One is taken from the petrophysical logs (not shown). The depth to the top of Core 3, Box 1, was measured as 6630′ (2020.82 m) at the time of drilling, and the base of Box 11 had a depth of 6661′6″ (2030.43 m). Each Box was subsequently given a depth in feet based on the length of the measured core (left scale in Supplementary Fig. S2). During storage, the cores expanded by up to 15%. Therefore, a third depth scale was constructed (in 2013 and updated in 2015) based on measuring the current core lengths, which is used in all figures and tables. The top of the core in Box 1 was taken as 2021.065 m, and all depths are appended below this in measured m of the current cores in Boxes 1–11 (right-hand scale in Supplementary Fig. S2). Deviation from the original core depth increases downwards due to the accumulation of expanded core.

The age model for E−8X is constructed with biostratigraphy and carbon isotope stratigraphy. *Axiodinium (Apectodinium) augustum*, a planktonic dinoflagellate cyst species used as the diagnostic marker for the PETM in the North Sea and Arctic[12,26,43,54], is present within the CIE of sediment core E−8X (Extended Data Table 1). The CIE main phase has been previously estimated to have lasted ~90 kyear (ref. [55]) or ~135 kyear (ref. [56]). The age models we use for Fig. 6 are those of Svalbard core BH9/05 'options A and B' (Fig. 3), constructed by cyclostratigraphic correlation[40]. We correlate the E−8X record (and other records used in Fig. 6) to BH9/05 using the δ$^{13}$C onset of the 'main phase' of the CIE, and the end of the 'recovery phase 1' along with the overall shape (Fig. 3). We favour option A as it provides an overall duration of the CIE of ~100 kyear, similar to the age model of (ref. [55]). To correlate 22/10a−4 with E−8X—a site some 350 km distant—we use a tie point at the onset of the CIE, and another at the beginning of the CIE main phase (dashed lines in Fig. 2). This coincides with an increase in TOC and onset of laminations at both sites, which is consistent with an anoxic water column from warmer water, elevated carbon flux, and a shift towards low salinity/eutrophic dinoflagellate cysts in 22/10a−4 (ref. [26]) (Fig. 2d). The average sedimentation rates for E−8X (assuming linear sedimentation rate) are 3 cm kyear$^{-1}$ (BH9/05 option A) or 2.1 cm kyear$^{-1}$ (BH9/05 option B).

Our records from E−8X indicate that the CIE onset took place over about 10 cm, which could be within 3–5 kyears. The largest shift in the δ$^{13}$C occurred over 6 cm or 2–3 kyear. Although the CIE onset has been previously estimated as lasting ~20 kyear from modelling Svalbard core BH9/05, the assumption that BH9/05 records the true onset length has been challenged[39] as that location was a marginal marine and proximal to land[57], and possibly impacted by terrestrial organic carbon lag times.

**TOC and carbon isotopes**. We analysed TOC and δ$^{13}$C$_{TOC}$ at high resolution (~1 cm) from sediment core E−8X (Figs. 2e, f), split for the first time in 2013. TOC analysis was performed on bulk samples by combustion in a Costech ECS4010 elemental analyser (EA) calibrated against an Acetanilide standard (Supplementary Data 1). Replicate analysis of well-mixed samples indicated a precision of ±<0.1. Carbon isotope analysis was carried out on bulk rock samples (Supplementary Data 1) by crushing the rock fragments using a ball mill. Any calcite was removed by placing the samples in 5% HCl overnight before rinsing with deionized water and drying down. $^{13}$C/$^{12}$C analyses were performed by combustion in a Costech EA on-line to a VG TripleTrap and Optima dual-inlet mass spectrometer, with δ$^{13}$C values calculated to the VPDB scale using a within-run laboratory standards calibrated against NBS-18, NBS-19 and NBS-22. Replicate $^{13}$C/$^{12}$C analyses were carried out on the section, and the mean standard deviation on the replicate analyses is <0.4‰. The δ$^{13}$C values show low scatter, and range between −24.7 and −31.7‰ (Supplementary Data 1). A proposed phytoplankton source for the organic carbon is consistent with the central North Sea position of E−8X, fine grained sedimentology, and a similar magnitude CIE when compared with δ$^{13}$C measured on dinoflagellate cysts (δ$^{13}$C$_{DINO}$) at Bass River (Fig. 3).

**Sedimentary Hg**. Hg analysis was carried out on bulk sediment samples by an RA-915 Portable Mercury Analyzer with PYRO-915 Pyrolyzer, Lumex, at the Department of Earth Sciences, University of Oxford. Methods were adapted from previous studies[58,59]. Approximately 50–100 mg of rock powder (depending on Hg enrichment) were measured into a glass measuring boat and its precise mass determined. Samples were heated in the Pyrolyzer to ~700 °C to volatilise the Hg within the sample. Following this, gaseous Hg was transported into the Analyzer and abundance was measured, providing the abundance mass of Hg as parts per billion (Supplementary Data 2). The machine was initially calibrated using six measurements of standard NIST 2587 with a Hg concentration of 290 ± 9 ppb and varying masses between 20 and 80 mg. During analysis, standards were analysed after every ten rock samples to ensure continuity of the calibration. The standard deviation of the standard was 31.1 ppb ($n = 82$), and the standard deviation of the offset from repeated samples from 22/10a−4 and E−8X ($n = 72$) was 26.7 ppb. For both records, sedimentary Hg shows highly fluctuating values from 6 to 1500 ppb (Fig. 2). An organic carbon association for this sedimentary Hg is supported by the relationships between Hg and TOC in 22/10a−4 and E−8X (Fig. 4), and the very low Hg values in samples with low TOC.

**Palynology**. Palynology samples were prepared using standard preparation procedures[60]. Samples were demineralised with hydrochloric (HCl) and hydrofluoric (HF) acids, and zinc bromide was used as a heavy liquid to separate and remove acid-resistant mineral grains. Slides were mounted using Elvacite. All palynomorphs were analyzed with a Nikon transmitted light microscope, counting the total number of palynomorphs on a strew slide (Extended Data Table 1).

## Data availability
The data that support the findings of this study are available within the Supplementary Information.

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

## Acknowledgements

This work was supported by NERC Isotope Geoscience Steering Committee (NIGFSC) Grants IP-1547-0515 and IP-1915-0619 (to S.K.), a European Research Council Consolidator Grant no. ERC-2018-COG-818717-V-ECHO (to T.A.M.), and forms part of a PhD project by E.M. funded by the College of Engineering, Mathematical and Physical Sciences, University of Exeter. Thanks to J. Boserup (GEUS) for consolidating and cutting core E−8X, and to L. Percival and F. Palmeri (University of Oxford) for sample analyses. M.J.L. and J.B.R. publish with the approval of the Executive Director, British Geological Survey (NERC).

## Author contributions

S.K. designed the study and analysed the data, K.D., S.K., J.B.R., K.B. and G.K.P. carried out sediment core sampling, G.K.P. carried out sediment logging, M.J.L. carried out isotope and organic carbon analysis, T.A.M. and E.M. carried out Hg analysis. S.K., K.B., G.K.P., K.D., T.A.M., E.M., A.R., J.B.R., T.W., S.P.H. and M.J.L. contributed to the writing of the paper.

## Competing interests

The authors declare no competing interests.
