## [Peer Review File · Nature Communications]

Paleocene/Eocene carbon feedbacks triggered by volcanic activityREVIEWER COMMENTS

Reviewer #1 (Remarks to the Author):

Kender et al. performed high-resolution Hg and TOC analysis on two North Sea sedimentary cores, and found anomalous high Hg concentration and Hg/TOC ratios preceding and within the early PETM, and conclude that volcanism from the North Atlantic Igneous Province was likely the trigger for the PETM. Given Hg concentration and Hg/TOC ratios are newly developed proxies of large volcanisms in geological history, overall, the results of this study support the authors' main conclusions. This paper is well organized and should be of interest to the readership of Nature Communications. I, Runsheng Yin, would like to recommend accepting this paper for publication in Nature Communications after minor revisions.

My major concern is how volcanic Hg entered the ocean during PETM. Large volcanism released large amounts of Hg into the atmosphere, which is globally transported and deposited to both terrestrial and oceanic systems. The authors seem to attribute the observed Hg anomalies to direct atmospheric deposition of volcanic Hg into the ocean. However, this may not be completely right, considering large volcanism also released massive CO₂, triggering global warming and chemical weathering, which would also promote the terrestrial Hg runoff. To solve the Hg source problems, I suggest the authors conduct Hg isotope analysis in their future studies. Mercury (Hg) isotopes are an effective source tracer in marine sediments (Grasby et al., 2017, 2019). Hg isotopes undergo both significant mass-dependent fractionation (MDF, represented by $\delta^{202}\text{Hg}$) and mass-independent fractionation (MIF, represented by $\Delta^{199}\text{Hg}$ and $\Delta^{201}\text{Hg}$) during Hg cycling. Hg-MDF occurs during various physical, chemical, and biological processes. Hg-MIF occurs mainly during photochemical processes (Blum et al., 2014) and is largely resistant to post-depositional alteration, providing clear source information. Volcanic emission is the predominant natural source of Hg, with $\Delta^{199}\text{Hg} \sim 0$ (Zambardi et al., 2009); however, photochemical processes alter the MIF signals during the global Hg transportation, resulting in positive $\Delta^{199}\text{Hg}$ in the oceanic pool (e.g., seawater) and negative $\Delta^{199}\text{Hg}$ in the terrestrial pool (e.g., vegetation and soil) (Blum et al., 2014). Negative shifts in $\Delta^{199}\text{Hg}$ are observed in marine sediments during mass extinction and ocean anoxic events related to enhanced soil erosion (Grasby et al., 2017; 2019; Them et al., 2019).

Minor comments:

Line 68: This is a wrong statement. Terrestrial Hg runoff can contribute large amounts of Hg into the ocean. Mercury enters the ocean primarily through direct atmospheric deposition and watershed runoff of land-based Hg sources (Sunderland and Mason, 2007; Amos et al., 2014; Zhang et al., 2015). In modern, globally, atmospheric Hg deposition (10 to 29 Mmol yr⁻¹) is in the same magnitude as input by watershed runoff (28±13 Mmol yr⁻¹). However, atmospheric deposition represents the major input of Hg to open oceans (Sunderland and Mason, 2007), whereas watershed-derived Hg is predominately deposited in margin seas (Amos et al., 2014; Zhang et al., 2015; Yin et al., 2015).

References

Amos, H. M., D. J. Jacob, D. Kocman, H. M. Horowitz, Y. Zhang, S. Dutkiewicz, M. Horvat, E. S. Corbitt, D. P. Krabbernhof, and E. M. Sunderland (2014), Global biogeochemical implications of mercury discharges from rivers and sediment burial. *Environ. Sci. Technol.*, 48(16), 9514-9522.

Blum, J.D., and Bergquist, B.A., 2007, Reporting of variations in the natural isotopic composition of mercury: *Analytical and Bioanalytical Chemistry*, v. 388, p. 353-359, <https://doi.org/10.1007/s00216-007-1236-9>.

Grasby, S.E., Shen, W., Yin, R., Gleason, J.D., Blum, J.D., Lepak, R.F., Hurley, J.P., and Beauchamp, B., 2017, Isotopic signatures of mercury contamination in latest Permian oceans: *Geology*, v. 45, p. 55-58, <https://doi.org/10.1130/G38487.1>.

Grasby, S.E., Them, T.R., Chen, Z.H., Yin, R.S., and Ardakani, O.H., 2019, Mercury as a proxy for volcanic emissions in the geologic record: *Earth-Science Reviews*, v. 196, p. 102880, <https://doi.org/10.1016/j.earscirev.2019.102880>.

Sunderland, E. M., D. P. Krabbernhof, J. W. Moreau, S. A. Strode, and W. M. Landing (2009). Mercury sources, distribution, and bioavailability in the North Pacific Ocean: Insights from data and models. *Global Biogeochem. Cy.*, 23(2).

Yin, R., X. Feng, B. Chen, J. Zhang, W. Wang, and X. Li (2015), Identifying the sources and processes of mercury in subtropical estuarine and ocean sediments using Hg isotopic composition. *Environ. Sci. Technol.*, 49(3), 1347-1355.

Zambardi, T., Sonke, J.E., Toutain, J.P., Sortino, F., and Shinohara, H., 2009, Mercury emissions and stable isotopic compositions at Vulcano Island (Italy): *Earth and Planetary Science Letters*, v. 277, p. 236–243, <https://doi.org/10.1016/j.epsl.2008.10.023>.

Zhang, Y., D. J. Jacob, S. Dutkiewicz, H. M. Amos, M. S. Long, and E. M. Sunderland (2015), Biogeochemical drivers of the fate of riverine mercury discharged to the global and Arctic oceans. *Global Biogeochem. Cy.*, 29(6): 854-864.

Reviewer #2 (Remarks to the Author):

The paper by Sev Kender and co-authors is a rigorous and interesting examination of the role of NAIP activity during the PETM using two North Sea localities. Overall, the manuscript is excellent and polished. The discussion and conclusions are a fair reflection of the data presented, and I believe it is worthy of publication in *Nature Communications*. I have a few suggestions that I believe would improve the paper, but these are easily achievable. Once these are addressed, I would recommend publication.

Line 208: I am not sure how useful it is to compare the surface carbon reservoirs of the late Permian with that of the Paleogene. Several factors were very different at 252 Ma, including global surface temperatures, plate tectonic configurations (Pangaea), and a lack of widespread pelagic photosynthetic organisms. Moreover, the fast carbon cycle had up the 3 times more carbon partitioned through the various surface reservoirs, so the magnitudes and rate of change to the end-Permian carbon cycle are not a great analogy for the Paleocene–Eocene transition.

Lines 212-219: The correlation with the temperature increase at Bass River prior to the start of the PETM is neglecting two studies from North Sea sediments in Denmark that show a marked temperature decrease across the same interval (Schoon et al., 2015; Stokke et al., 2020a). Temperature fluctuations prior to the PETM, and the role of volcanic activity in these changes, need to include the measurements proximal to the study area.

Lines 220-223: The change in Hg/TOC levels across the onset of the PETM may well be evidence of reduced volcanic activity, but that is not the only possible explanation. Given that several PETM sites show a large change in sedimentation rates across the PETM onset, coupled with a change in dominant source of TOC, both of these factors could alter the Hg/TOC ratio without any change to the level of volcanic activity. A little more discussion on other potential causes of low Hg/TOC across the PETM onset is warranted.

Lines 320-323: The recent work on the Stolleklint beach section would be good to cite here to corroborate the regional nature of seawater anoxia in the North Sea (Schoon et al., 2015; Stokke et al., 2020c).

Line 330: The most precise estimate for the thickness of the Stolleklint clay is 24.4 m at the type locality of Stolleklint (Jones et al., 2019). This would suggest that it was deposited in ~100 kyr rather than 60 kyr (based on the 15 m estimate), which also matches other estimations of the duration of the PETM body (e.g. van der Meulen et al., 2020; and references already cited in line 347).

Figure 1: Something seems to have gone wrong with the coastline of Denmark. Most of western Jylland is missing in the current figure.

Figure 2: Given the excellent preservation of ash layers in Danish sediments before, during and after the PETM CIE, and the presence of ash layers in the E-8X core, could more be done to use key horizons as marker beds to correlate between the two localities?

Figure 3: Most of the current astronomical solutions agree that the PETM onset began at 55.93 Ma

at the latest (e.g. Westerhold et al., 2017), with some even suggesting that it began earlier at 56.01 Ma (Zeebe and Lourens, 2019). An age of 55.822 Ma for the onset of the PETM is not compatible with published astronomical solutions.

Figure 5: I am not convinced by all the "Reduced volcanic Hg" labels. In Svalbard, there is a ramping up of the baseline of Hg/TOC across the PETM onset, such that the part labelled as reduced volcanic Hg is actually greater than much of the pre-PETM interval. At Fur the onset is contemporaneous with at least an order of magnitude increase in sediment deposition rates, which means that even though the Stolleklint clay has lower Hg/TOC values than pre-PETM strata, the rate of Hg deposition is likely to be significantly greater.

Suggested References:

van der Meulen, B., Gingerich, P. D., Lourens, L. J., Meijer, N., van Broekhuizen, S., van Ginneken, S., & Abels, H. A. (2020). Carbon isotope and mammal recovery from extreme greenhouse warming at the Paleocene–Eocene boundary in astronomically-calibrated fluvial strata, Bighorn Basin, Wyoming, USA. *Earth and Planetary Science Letters*, 534, 116044. <https://doi.org/10.1016/j.epsl.2019.116044>.

Schoon, P. L., Heilmann-Clausen, C., Schultz, B. P., Sinninghe-Damsté, J. S., & Schouten, S. (2015). Warming and environmental changes in the eastern North Sea Basin during the Palaeocene–Eocene Thermal Maximum as revealed by biomarker lipids. *Organic Geochemistry*, 78, 79-88. <https://doi.org/10.1016/j.orggeochem.2014.11.003>.

Stokke, E. W., Jones, M. T., Tierney, J. E., Svensen, H. H., & Whiteside, J. H. (2020a). Temperature changes across the Paleocene-Eocene Thermal Maximum—a new high-resolution TEX86 temperature record from the Eastern North Sea Basin. *Earth and Planetary Science Letters*, 544, 116388. <https://doi.org/10.1016/j.epsl.2020.116388>.

Stokke, E. W., Jones, M. T., Riber, L., Haflidason, H., Midtkandal, I., Schultz, B. P., & Svensen, H. H. (2020c). Rapid and sustained environmental responses to global warming: The Paleocene–Eocene Thermal Maximum in the eastern North Sea. *Climate of the Past Discussions*, 1-38. <https://doi.org/10.5194/cp-2020-150>.

Westerhold, T., Röhl, U., Frederichs, T., Agnini, C., Raffi, I., Zachos, J. C., & Wilkens, R. H., (2017). Astronomical calibration of the Ypresian timescale: implications for seafloor spreading rates and the chaotic behavior of the solar system?: *Climate of the Past* 13 (9), 1129-1152.

Zeebe, R. E., & Lourens, L. J. (2019). Solar System chaos and the Paleocene–Eocene boundary age constrained by geology and astronomy. *Science* 365 (6456), 926-929.

RESPONSE TO REVIEWER COMMENTS

Reviewer #1:

Kender et al. performed high-resolution Hg and TOC analysis on two North Sea sedimentary cores, and found anomalous high Hg concentration and Hg/TOC ratios preceding and within the early PETM, and conclude that volcanism from the North Atlantic Igneous Province was likely the trigger for the PETM. Given Hg concentration and Hg/TOC ratios are newly developed proxies of large volcanisms in geological history, overall, the results of this study support the authors' main conclusions. This paper is well organized and should be of interest to the readership of Nature Communications. I, Runsheng Yin, would like to recommend accepting this paper for publication in Nature Communications after minor revisions.

Response: We thank Runsheng Yin for recognising the importance of the work, and for the time taken to review the paper.

My major concern is how volcanic Hg entered the ocean during PETM. Large volcanism released large amounts of Hg into the atmosphere, which is globally transported and deposited to both terrestrial and oceanic systems. The authors seem to attribute the observed Hg anomalies to direct atmospheric deposition of volcanic Hg into the ocean. However, this may not be completely right, considering large volcanism also released massive CO₂, triggering global warming and chemical weathering, which would also promote the terrestrial Hg runoff. To solve the Hg source problems, I suggest the authors conduct Hg isotope analysis in their future studies. Mercury (Hg) isotopes are an effective source tracer in marine sediments (Grasby et al., 2017, 2019). Hg isotopes undergo both significant mass-dependent fractionation (MDF, represented by $\delta^{202}\text{Hg}$) and mass-independent fractionation (MIF, represented by $\Delta^{199}\text{Hg}$ and $\Delta^{201}\text{Hg}$) during Hg cycling. Hg-MDF occurs during various physical, chemical, and biological processes. Hg-MIF occurs mainly during photochemical processes (Blum et al., 2014) and is largely resistant to post-depositional alteration, providing clear source information. Volcanic emission is the predominant natural source of Hg, with $\Delta^{199}\text{Hg} \sim 0$ (Zambardi et al., 2009); however, photochemical processes alter the MIF signals during the global Hg transportation, resulting in positive $\Delta^{199}\text{Hg}$ in the oceanic pool (e.g., seawater) and negative $\Delta^{199}\text{Hg}$ in the terrestrial pool (e.g., vegetation and soil) (Blum et al., 2014). Negative shifts in $\Delta^{199}\text{Hg}$ are observed in marine sediments during mass extinction and ocean anoxic events related to enhanced soil erosion (Grasby et al., 2017; 2019; Them et al., 2019).

Response: We thank Runsheng Yin for this interesting and useful information and will strongly consider measuring Hg isotopes in the future if time and resources permit. Please also see our response to the next point.

Minor comments:

Line 68: This is a wrong statement. Terrestrial Hg runoff can contribute large amounts of Hg into the ocean. Mercury enters the ocean primarily through direct atmospheric deposition and watershed runoff of land-based Hg sources (Sunderland and Mason, 2007; Amos et al., 2014; Zhang et al., 2015). In modern, globally, atmospheric Hg deposition (10 to 29 Mmol yr⁻¹) is in the same magnitude as input by watershed runoff (28±13 Mmol yr⁻¹). However, atmospheric deposition represents the major input of Hg to open oceans (Sunderland and Mason, 2007), whereas watershed-derived Hg is predominately deposited in margin seas

(Amos et al., 2014; Zhang et al., 2015; Yin et al., 2015).

Response: We thank Runsheng Yin for this and have emended the text accordingly:

“Although modern dissolved Hg input from freshwater runoff to oceans is substantial ($27\pm 13 \text{ Mmol yr}^{-1}$)²⁷ compared to atmospheric deposition ($19.5\pm 9.5 \text{ Mmol yr}^{-1}$)²⁸, isotopic studies have shown terrestrial Hg to be constrained to coastal and nearshore shelf environments²⁹ whilst our core sites are >200 km distal in the central North Sea.”
Lines 66-70

Yin et al. (2015) show very clearly in their impressive study of the Pearl River Estuary and adjacent South China Sea that Hg isotopes indicate riverine input of Hg to deep water sediments was important less than 50 km from the river mouth, with more distal locations appearing to show much reduced influence. We suggest that our sites are less likely to be heavily affected by terrigenous Hg as they are >200 km from the nearest landmasses, and are not directly collected from a large delta system.

References

- Amos, H. M., D. J. Jacob, D. Kocman, H. M. Horowitz, Y. Zhang, S. Dutkiewicz, M. Horvat, E. S. Corbitt, D. P. Krabbernhof, and E. M. Sunderland (2014), Global biogeochemical implications of mercury discharges from rivers and sediment burial. *Environ. Sci. Technol.*, 48(16), 9514-9522.
- Blum, J.D., and Bergquist, B.A., 2007, Reporting of variations in the natural isotopic composition of mercury: *Analytical and Bioanalytical Chemistry*, v. 388, p. 353–359, <https://doi.org/10.1007/s00216-007-1236-9>.
- Grasby, S.E., Shen, W., Yin, R., Gleason, J.D., Blum, J.D., Lepak, R.F., Hurley, J.P., and Beauchamp, B., 2017, Isotopic signatures of mercury contamination in latest Permian oceans: *Geology*, v. 45, p. 55–58, <https://doi.org/10.1130/G38487.1>.
- Grasby, S.E., Them, T.R., Chen, Z.H., Yin, R.S., and Ardakani, O.H., 2019, Mercury as a proxy for volcanic emissions in the geologic record: *Earth-Science Reviews*, v. 196, p. 102880, <https://doi.org/10.1016/j.earscirev.2019.102880>.
- Sunderland, E. M., D. P. Krabbernhof, J. W. Moreau, S. A. Strode, and W. M. Landing (2009). Mercury sources, distribution, and bioavailability in the North Pacific Ocean: Insights from data and models. *Global Biogeochem. Cy.*, 23(2).
- Yin, R., X. Feng, B. Chen, J. Zhang, W. Wang, and X. Li (2015), Identifying the sources and processes of mercury in subtropical estuarine and ocean sediments using Hg isotopic composition. *Environ. Sci. Technol.*, 49(3), 1347-1355.
- Zambardi, T., Sonke, J.E., Toutain, J.P., Sortino, F., and Shinohara, H., 2009, Mercury emissions and stable isotopic compositions at Vulcano Island (Italy): *Earth and Planetary Science Letters*, v. 277, p. 236–243, <https://doi.org/10.1016/j.epsl.2008.10.023>.
- Zhang, Y., D. J. Jacob, S. Dutkiewicz, H. M. Amos, M. S. Long, and E. M. Sunderland (2015), Biogeochemical drivers of the fate of riverine mercury discharged to the global and Arctic oceans. *Global Biogeochem. Cy.*, 29(6): 854-864.

Reviewer #2:

The paper by Sev Kender and co-authors is a rigorous and interesting examination of the role of NAIP activity during the PETM using two North Sea localities. Overall, the manuscript is

excellent and polished. The discussion and conclusions are a fair reflection of the data presented, and I believe it is worthy of publication in Nature Communications. I have a few suggestions that I believe would improve the paper, but these are easily achievable. Once these are addressed, I would recommend publication.

Response: We thank the reviewer for their assessment of our work, and for the time taken to review the paper.

Line 208: I am not sure how useful it is to compare the surface carbon reservoirs of the late Permian with that of the Paleogene. Several factors were very different at 252 Ma, including global surface temperatures, plate tectonic configurations (Pangaea), and a lack of widespread pelagic photosynthetic organisms. Moreover, the fast carbon cycle had up the 3 times more carbon partitioned through the various surface reservoirs, so the magnitudes and rate of change to the end-Permian carbon cycle are not a great analogy for the Paleocene–Eocene transition.

Response: We fully agree with the reviewer and have now removed this part of the sentence from the text.

Lines 212-219: The correlation with the temperature increase at Bass River prior to the start of the PETM is neglecting two studies from North Sea sediments in Denmark that show a marked temperature decrease across the same interval (Schoon et al., 2015; Stokke et al., 2020a). Temperature fluctuations prior to the PETM, and the role of volcanic activity in these changes, need to include the measurements proximal to the study area.

Response: We agree and now cite and discuss these two SST records within the text:

“SST records from nearby Fur^{43,44} are slightly more complicated to interpret due to occasionally high branched and isoprenoid tetraether (BIT) index values and changing sedimentation rates making it harder to correlate, but do show a possible fall stratigraphically below the CIE that has been suggested to reflect local cooling from volcanism⁴⁴.” Lines 238-242

Lines 220-223: The change in Hg/TOC levels across the onset of the PETM may well be evidence of reduced volcanic activity, but that is not the only possible explanation. Given that several PETM sites show a large change in sedimentation rates across the PETM onset, coupled with a change in dominant source of TOC, both of these factors could alter the Hg/TOC ratio without any change to the level of volcanic activity. A little more discussion on other potential causes of low Hg/TOC across the PETM onset is warranted.

Response: We now cite and discuss the possibility of changing TOC sources within the text as suggested:

“These Hg/TOC pulses are not likely to have been caused by changing sedimentation rates, as reporting Hg as a ratio to TOC aims to remove the influence of changing background sedimentation. However, changing delivery and source of sedimentary TOC (transportation and reworking) can modify Hg/TOC trends.” Lines 149-152

And:

“Some records are thought to be influenced by sediment reworking (e.g., Svalbard and 22/10a-4) that could reduce Hg/TOC signals, but E-8X shows no sign of reworking with a clear and rapid CIE onset and central deep North Sea Basin location.” Lines 206-209

Lines 320-323: The recent work on the Stolleklint beach section would be good to cite here to corroborate the regional nature of seawater anoxia in the North Sea (Schoon et al., 2015; Stokke et al., 2020c).

Response: This is a good suggestion and we have now cited Stokke et al. (2020).

Line 330: The most precise estimate for the thickness of the Stolleklint clay is 24.4 m at the type locality of Stolleklint (Jones et al., 2019). This would suggest that it was deposited in ~100 kyr rather than 60 kyr (based on the 15 m estimate), which also matches other estimations of the duration of the PETM body (e.g. van der Meulen et al., 2020; and references already cited in line 347).

Response: Many thanks for pointing this out, we have now emended this section in the text as suggested.

Figure 1: Something seems to have gone wrong with the coastline of Denmark. Most of western Jylland is missing in the current figure.

Response: We have now corrected the figure to include a much more accurate and complete outline of Denmark.

Figure 2: Given the excellent preservation of ash layers in Danish sediments before, during and after the PETM CIE, and the presence of ash layers in the E-8X core, could more be done to use key horizons as marker beds to correlate between the two localities?

Response: Although we do agree there may be ashes that correlate between E-8X and onshore Danish sections such as at Fur Island (Stokke et al. 2020), we are cautious as we have no geochemical data from these ash beds to support any proposed correlation. As the two sites are far apart (~300 km) we think it is possible that there are some ashes not represented at both sites. It is clear that both sites have 2–3 ash beds just before the CIE onset, have very few or no discernible ashes in the Main Phase, and then have several ash beds during the Recovery Phase 1 (5 at E-8X and 4 at Fur). We propose not to discuss this point as it is not essential to the paper, but will include it in the text or the figure caption if the reviewer and/or editor recommend.

Figure 3: Most of the current astronomical solutions agree that the PETM onset began at 55.93 Ma at the latest (e.g. Westerhold et al., 2017), with some even suggesting that it began earlier at 56.01 Ma (Zeebe and Lourens, 2019). An age of 55.822 Ma for the onset of the PETM is not compatible with published astronomical solutions.

Response: We agree that the absolute ages in Charles et al. (2011) are out of date, and have therefore removed them from Fig. 3 and used a relative age scale instead. The important aspect is the amount of time encapsulated within the section, based on relative orbital tuning. We have now cited Zeebe & Lourens (2019) in the introduction when referring to the age of the PETM, and orbital variations.

Figure 5: I am not convinced by all the “Reduced volcanic Hg” labels. In Svalbard, there is a ramping up of the baseline of Hg/TOC across the PETM onset, such that the part labelled as reduced volcanic Hg is actually greater than much of the pre-PETM interval. At Fur the onset is contemporaneous with at least an order of magnitude increase in sediment deposition rates, which means that even though the Stollekint clay has lower Hg/TOC values than pre-PETM strata, the rate of Hg deposition is likely to be significantly greater.

Response: In order to avoid any ambiguity we have now removed the labels and arrows, and instead describe in the figure caption what we are aiming to highlight.

“Carbon isotope excursion (CIE) step 2 is shown as a dashed line, and does not co-occur with an Hg or Hg/TOC spike in the sections.” Lines 667-668

Although an increase in sedimentation rate would not alter Hg/TOC (as both would be diluted), we agree that absolute Hg would be reduced, and that hypothetical reworked older carbon, or a changing source, could change the values. Our main point is that there is no large Hg/TOC spike at any of the locations during CIE step 2, which is what a volcanic trigger hypothesis for this carbon would produce. We have therefore also added the following to the text:

“In Svalbard, there is a general increase in the Hg/TOC baseline over the CIE onset, and at Fur sedimentation rates significantly increase during the PETM⁴¹ such that Hg deposition rates likely increase even though concentrations do not. However, there is no Hg/TOC evidence for any substantial increase in volcanism during CIE step 2 (unlike step 1).” Lines 209-213

Suggested References:

van der Meulen, B., Gingerich, P. D., Lourens, L. J., Meijer, N., van Broekhuizen, S., van Ginneken, S., & Abels, H. A. (2020). Carbon isotope and mammal recovery from extreme greenhouse warming at the Paleocene–Eocene boundary in astronomically-calibrated fluvial strata, Bighorn Basin, Wyoming, USA. *Earth and Planetary Science Letters*, 534, 116044. <https://doi.org/10.1016/j.epsl.2019.116044>.

Schoon, P. L., Heilmann-Clausen, C., Schultz, B. P., Sinninghe-Damsté, J. S., & Schouten, S. (2015). Warming and environmental changes in the eastern North Sea Basin during the Palaeocene–Eocene Thermal Maximum as revealed by biomarker lipids. *Organic Geochemistry*, 78, 79-88. <https://doi.org/10.1016/j.orggeochem.2014.11.003>.

Stokke, E. W., Jones, M. T., Tierney, J. E., Svensen, H. H., & Whiteside, J. H. (2020a). Temperature changes across the Paleocene-Eocene Thermal Maximum—a new high-resolution TEX86 temperature record from the Eastern North Sea Basin. *Earth and Planetary Science Letters*, 544, 116388. <https://doi.org/10.1016/j.epsl.2020.116388>.

Stokke, E. W., Jones, M. T., Riber, L., Haflidason, H., Midtkandal, I., Schultz, B. P., & Svensen, H. H. (2020c). Rapid and sustained environmental responses to global warming: The Paleocene–Eocene Thermal Maximum in the eastern North Sea. *Climate of the Past Discussions*, 1-38. <https://doi.org/10.5194/cp-2020-150>.

Westerhold, T., Röhl, U., Frederichs, T., Agnini, C., Raffi, I., Zachos, J. C., & Wilkens, R. H., (2017). Astronomical calibration of the Ypresian timescale: implications for seafloor spreading rates and the chaotic behavior of the solar system?: *Climate of the Past* 13 (9), 1129-1152.

Zeebe, R. E., & Lourens, L. J. (2019). Solar System chaos and the Paleocene–Eocene boundary age constrained by geology and astronomy. *Science* 365 (6456), 926-929.